# Assessing the Removal of Arsenite and Arsenate Mixtures from the Synthetic Bangladesh Groundwater (SBGW) Using Combined Fe(VI)/Fe(III) Treatments and Local Regression Analysis

Javier Quino-Favero [1],*[image: ORCID], Raúl Eyzaguirre Perez [1], Patricia Prieto Veramendi [1][image: ORCID], Paloma Mogrovejo García [1] and Lisveth Flores del Pino [2]

[1]  Research Group of Technological Solutions for the Environment, Faculty of Engineering and Architecture, Universidad de Lima, Lima 33, Peru; reyzagui@ulima.edu.pe (R.E.P.); paty.mpv@gmail.com (P.P.V.); mogrovejopaloma@gmail.com (P.M.G.)

[2]  Center for Environmental Research in Chemistry, Toxicology and Biotechnology, Chemistry Department, Faculty of Science, Universidad Nacional Agraria La Molina, Lima 33, Peru; lisveth@lamolina.edu.pe

*  Correspondence: jquinof@ulima.edu.pe

**Abstract:** Arsenic is an inorganic pollutant that, depending on oxidation–reduction and pH level conditions, may be found in natural waters in two variants: As(III) and As(V). Any treatment to effectively remove arsenic from water will be conditioned by the presence of one or both variants. In this context, this study assesses using electrochemically produced Fe(VI) with Fe(III) to remove As(III), As(V), and their combinations from the Synthetic Bangladesh Groundwater (SBGW) containing anions that interfere with iron-based arsenic removal processes. The combined use of Fe(VI) and Fe(III) allowed us to remove the total arsenic below the 10 mg $L^{-1}$ threshold established by the World Health Organization and Peruvian regulations for drinking water. An optimum combination of 1 mg $L^{-1}$ of Fe(VI) and 30 mg $L^{-1}$ of Fe(III) was identified and tested on the removal of four different proportions of As(III):As(V) for two total concentrations: 500 and 250 mg $L^{-1}$. There were no significant differences in the final removal values under the different proportions of As(III):As(V) for each total concentration, with a final removal average of 99.0% and 96.9% for the 500 and 250 μg $L^{-1}$ concentrations, respectively.

**Keywords:** Ferrate(VI); arsenite; arsenate; groundwater

## 1. Introduction

Arsenic is a metalloid widely distributed throughout the Earth's crust, and it is released into water sources as part of a leaching process from rocks and sediments, as well as from anthropogenic sources [1]. Freshwater arsenic concentrations may range from minor traces to as high as 44,000 μg $L^{-1}$ in Waiotapu Valley, New Zealand [2]. Considering the 10 μg $L^{-1}$ threshold proposed by the World Health Organization (WHO), 100 million people around the world are at risk of arsenic exposure from drinking water. In particular, 45 million people in Asian developing countries are exposed to concentrations exceeding 45 μg $L^{-1}$ [3]. Therefore, the WHO considers that contaminated drinking water poses the greatest threat to public health from arsenic [4].

Arsenic groundwater contamination has had significant negative impacts on human health, with the arsenic poisoning cases reported in Bangladesh and West Bengal being the most prominent examples [5]. The toxic effects of arsenic in adults, depending on exposure levels and times, include skin injuries, cardiovascular effects, gastrointestinal disturbances, liver disease, and cancer [6]. In children, lung disease and defective intellectual functions [7] are its toxic effects. Now, As(III) is reported as being 25–60 times more toxic than As(V) and hundreds of times more toxic than its methylated forms [8,9]. In oral epidermal carcinoma

cell lines, As(V) uptake is associated with diffusion, while As(III) uptake requires an energy-dependent transport system, similar to the one required in phosphate uptake. These differences represent an important factor in the elevated toxicity of As(III) [10]. The pathogenesis of arsenic-induced toxicity is also associated with the damage caused by Reactive Oxygen Species (ROS) [6].

In natural waters, arsenic is found as oxyanions or in neutral form and, depending on Eh and pH levels, it may be released as types of As(III) and As(V). As(III) exists as $H_3AsO_3$, but it can also be found as $H_2AsO_3^-$, $HAsO_3^{2-}$, and $AsO_3^{3-}$ ions; As(V) exists as $H_3AsO_4$, but it can also be found as $H_2AsO_4^-$, $HAsO_4^{2-}$, and $AsO_4^{3-}$ anions [11,12]. Both arsenic types can coexist as vertically distributed in groundwater [8,13], but the As(III) type is usually the prevailing one [14]. Hence, groundwater treatments must be capable of removing both arsenic species. As(V) can be removed from the water through coprecipitation with Fe(III), whereas using $FeCl_3$ to remove As(III) is more influenced by groundwater composition [15]. Thus, some studies focusing on the removal of arsenic from water use the Synthetic Bangladesh Groundwater as a model [16–19].

As(III) preoxidation is required for effectively removing arsenic from water samples [20]. It can be achieved using ozone, chlorine, hypochlorite, $H_2O_2$, or Fe(VI) [20,21].

Fe(VI) was used along with Fe(III) for removing As(III) from the Nakdong River waters, reducing the initial concentration of 517 µg $L^{-1}$ to under 50 µg $L^{-1}$ [22]. Other authors studied the use of Fe(VI) to remove an initial concentration of 500 µg $L^{-1}$ of As(III) from waters containing phosphate, silicate, and bicarbonate ions [23,24], achieving a final concentration of under 10 µg $L^{-1}$. Arsenic removal assays in the presence of some ions found in natural waters—such as phosphate, silicate, bicarbonate—may prevent the removal of arsenic [25] and are stringent tests. Groundwater usually contains combinations of As(III) and As(V) at different concentration levels. A study conducted in the province of Santa Fe, Argentina, reported that up to 36% of the total arsenic found in groundwater was in trivalent arsenic form [26] while another study in Paba Upazila of Rajshahi district, Bangladesh reported 88% for the same species [27]. Previous studies on arsenic removal have focused on a single species: As(III) [28–30], As(V) [31–33], or both species in independent assays [34,35]. As arsenic polluted groundwater contains both arsenic species and anions known to interfere with the removal, testing combined Fe(VI)/Fe(III) treatment in the presence of competing anions and varying ratios of both arsenic species could pinpoint potential limitations of the proposed treatment.

Few studies have evaluated arsenic removal methods when both inorganic arsenic species coexist in the same water. A study [29] using titanium xerogel coagulant reported that when As(III) coexisted in the same solution with As(V) the removal efficiency of As(V) was not affected, while As(III) removal decreased in a range from 4% to 17%, pointing out that arsenic species could interact and impair the efficiency of arsenic removal. Other groups studied simultaneous arsenic removal by adsorption of As(III) and As(V) using a photoactive selective adsorbent to promote the oxidation of As(III) to As(V) [36] or used a selective adsorbent for both arsenic species [37]. To our knowledge, this is the first time Fe(VI)/Fe(III) combined treatment is used to remove As(III)/As(V) mixtures in simulated underground water.

Therefore, this study aimed to: assess the effectiveness of Fe(VI)/Fe(III) combined treatment for removing arsenic when both ions, As(III) and As(V), were present simultaneously in the Synthetic Bangladesh Groundwater (SBGW);And assess the best Fe(VI)/Fe(III) combination with four As(III):As(V) ratios at two concentrations (250 and 500 µg $L^{-1}$) by measuring the residual concentration of each arsenic species at the end of the treatment.

## 2. Materials and Methods

### 2.1. Synthetic Bangladesh Groundwater (SBGW) Preparation

All arsenic removal tests were performed using the Synthetic Bangladesh Groundwater or SBGW [38] in which arsenite, arsenate, or both had been added. The SBGW was prepared using ultrapure water (18 MΩ·cm) and stock solutions of $Na_2HPO_4 \cdot 7H_2O$,

NaHCO$_3$, CaSO$_4$ ·2H$_2$O, MgCl$_2$ ·6H$_2$O, CaCl$_2$, and Na$_2$SiO$_3$·5H$_2$O (Table 1). In addition, the pH level of the water was adjusted to 7.0 ± 0.2 using carbon dioxide. The water composition used was similar to the one used in a previous study [19,38].

**Table 1.** SBGW composition used in this study.

| | PO$_4^{3-}$ | SiO$_3^{2-}$ | SO$_4^{2-}$ | Ca$^{2+}$ | Mg$^{2+}$ | Cl$^-$ | Na$^+$ | HCO$_3^-$ | Fe |
|---|---|---|---|---|---|---|---|---|---|
| Concentrations in mg L$^{-1}$ | 1.3 | 19.5 | 8 | 61 | 8 | 125 | 138 | 275 | 0 |

### 2.2. Ferrate(VI) Quantification

The amount of ferrate present in the water was quantified via visible spectroscopy using a Shimadzu UV-2600 spectrophotometer. Its concentration level was calculated according to Equation (1).

$$[FeO_4]^{2-} = \frac{\Delta_{Abs} \, V_{final}}{\varepsilon \, \ell \, V_{sample}} \tag{1}$$

where $\Delta_{Abs}$ is the absorbance difference from the corresponding blank measured at 505 nm, $V_{final}$ is the sum of sample volumes, $V_{sample}$ is the volume of solution added for dilution, $\varepsilon$ is the reported molar extinction coefficient [39] for Fe(VI) at 505 nm (1070 L mol$^{-1}$ cm$^{-1}$), and $\ell$ represents the cell width (1 cm).

### 2.3. Ferrate(VI) Electrochemical Synthesis

The electrochemical synthesis of ferrate was conducted in a polymethylmethacrylate (PMMA) cell comprised an anodic chamber with an iron electrode and a cathode chamber with a graphite electrode. Both chambers were separated by a cation exchange membrane (CTIEM-1 Perfluorosulfonic Acid Cation Exchange Membrane Zibo Cantian, China) at 2.3 Ω cm$^2$ (pore diameter < 100 nm according to FSEM measurements). The electrode area used was 25 cm$^2$ and the electrolyte was 20 mol L$^{-1}$ NaOH. The ferrate was generated at a current density of 80 A m$^{-2}$ for 5 h, with which a 0.28-mol L$^{-1}$ solution of Fe(VI) was achieved as described above [40].

### 2.4. Treatment Tests

A continuous variable-speed multiple flocculator (Platypus Jar Tester, Microfloc Pty) with 1-L square-section jars was used. The tests aimed at removing arsenic in its As(III) and As(V) forms, as well as a combination of both. At the beginning of the test, Fe(VI) was added. Then, Fe(III) was added while mixing rapidly at 200 rpm for 60 s (velocity gradient of 726 s$^{-1}$). After 60 s, mixing continued at a slow rate of 60 rpm for 15 min (speed gradient of 119.9 s$^{-1}$). After performing the jar test, the final reported pH level was 7.23 ± 0.15. Next, the samples were allowed to settle for 12 h, and an aliquot was taken from the supernatant to determine the total arsenic concentration. During the speciation tests, arsenic concentrations were determined immediately after treatment or using samples preserved with EDTA and acetic acid [41] to avoid any preanalysis oxidation of As(III).

For generating the local regression surfaces and the interaction analysis, each of the three tested arsenic concentrations (1000 μg L$^{-1}$ As(V); 1000 μg L$^{-1}$ As(III); 1000 μg L$^{-1}$ As(III)/As(V)) was confronted against a concentration of 15, 30, 45 and 60 mg L$^{-1}$ of Fe(III) ions from FeCl$_3$ and a concentration of 0, 0.5, 0.9, and 1.3 mg L$^{-1}$ Fe(VI) ions, resulting in 16 combinations per treatment. Subsequently, the removal of As(III) and As(V) combinations was evaluated in 500 and 250 μg L$^{-1}$ concentrations at 80:20, 60:40, 40:60, and 20:80 proportions of As(III) and As(V) and using the combination of Fe(VI) and Fe(III) that achieved the best removal results for both species (1 mg Fe(VI) and 30 mg Fe(III)).

### 2.5. Determination of Total Concentrations of Arsenic and its Species

Total arsenic analyses required a prereduction step to convert all As(V) to As(III) [42], which generated AsH$_3$. Then, 12.5 mL of concentrated HCl and 1 mL of potassium iodide–ascorbic acid reducing solutions were added to a 25-mL sample. After mixing, the sample

was left for 30 min to complete the reaction, before being brought to a final volume of 50 mL with ultrapure water. Next, we took 2.5 mL from the said sample and analyzed it using the PSA Millennium Excalibur HG-AFS. The reducing KI/ascorbic acid solution was prepared by dissolving 25 g of KI and 5 g of ascorbic acid in 50 mL of ultrapure water. The blank/carrier was prepared by mixing 250 mL of concentrated HCl and 20 mL of the KI/ascorbic acid solution, which was then brought to a volume of 1 L. To determine As(III) and As(V) concentration levels, 250 μL of each sample was introduced into a chromatographic column (Hamilton, PRP-X100 10 μm, 4.1 × 250 mm) with a flow rate of 0.7 mL min$^{-1}$ at 650 psi. The mobile phase used for the separation was a $NaH_2PO_4$–$Na_2HPO_4$ 20-mM buffer at a pH level of 6.20, which was degassed and filtered in 0.2 μm (Merck Millipore Durapore membrane filter GVWP04700). After the chromatographic separation, the arsenic species were analyzed using the PSA Millennium Excalibur HG-AFS. Detection limits for As(III) and As(V) are 0.1 and 0.2 μg L$^{-1}$, respectively.

### 2.6. Statistical Analysis

Two approaches were used to assess the effects of Fe(VI) and Fe(III) on arsenic removal. First, by means of the quadratic polynomial regression model with interaction (Equation (2)):

$$y = \beta_0 + \beta_1 x_1 + \beta_2 x_1^2 + \beta_3 x_2 + \beta_4 x_2^2 + \beta_5 x_1 x_2 + e, \tag{2}$$

where $y$ is the final concentration of arsenic, $x_1$ is the Fe(III) concentration, $x_2$ is the Fe(VI) concentration, and $e$ is a normally distributed error term. Through this model, we assessed the significance of the linear, quadratic, and interaction effects of the Fe(VI) and Fe(III) concentration levels in the removal of arsenic. Second, with a local regression model fitted using quadratic polynomials in $x_1$ and $x_2$. With this method, at each $x$ point of the regressor domain, the model is adjusted using the neighboring points, weighted by their distance from $x$, to obtain a smooth representation of the relationship between the regression variables and the response. This fitted response surface is more flexible than that obtained in model (2) because it does not assume the form of a polynomial function over the entire regression variable domain.

Differences between the remotion of 500 and 250 μg L$^{-1}$ concentrations of As(III) and As(V) at 80:20, 60:40, 40:60, and 20:80 proportions with a fixed concentration of Fe(III) and Fe(VI) were evaluated with an analysis of variance for the two factors, concentrations and proportions.

All statistical analysis was done using the R software [43]. The local regression model implemented in R follows Cleveland et al. [44].

### 3. Results and Discussion

#### 3.1. Effects of Ferrate(VI) Ions Combined with Fe(III) Ions in the Removal of Arsenic(III)

Table 2 shows significant effects for all model terms. The sign of the linear term coefficients is negative; this implies that the increase in Fe(III) and Fe(VI) concentrations produces lower final As(III) concentrations. This behavior can also be seen in Figure 1, where the lowest As(III) concentration values are displayed in the upper right corner. The interaction, with a positive coefficient value, suggests a competition effect; in Figure 1 we see that as Fe(VI) increases, the marginal effect of Fe(III) decreases. This effect is consistent with Prucek et al. [45] which demonstrated that Fe(VI) alone could be used to achieve simultaneous As(III) oxidation and coprecipitation. Fe(VI) oxidizes As(III) to As(V) (Equation (3)), which is in turn adsorbed and coprecipitated by Fe(III) ions undergoing hydrolysis to form iron oxyhydroxides (Equation (4)), iron arsenate precipitate could be formed to some extent, but this complexation reaction (Equation (5)) is weak under neutral and alkaline conditions [30].

$$2HFeO_4^- + 3AsO_3^{-3} + 3H_2O \rightarrow 2\,Fe^{3+} + 3AsO_4^{-3} + 8\,OH^- \tag{3}$$

$$Fe^{+3} + 3H_2O \rightarrow Fe\,(OH)_3 + 3H^+ \tag{4}$$

$$AsO_4^{-3} + Fe^{3+} \rightarrow FeAsO_4 \tag{5}$$

**Table 2.** *t*-test coefficients for a Quadratic Polynomial Regression Model with interaction that explains the final arsenic concentration at the initial As(III) concentration of 1 mg $L^{-1}$.

| Factors | Estimated Coefficient | Standard Error | *t*-Value | *p*-Value |
|---|---|---|---|---|
| Fe(III) | −0.017683 | 0.002826 | −6.258 | 0.0001 |
| Fe(VI) | −0.684623 | 0.068568 | −9.985 | 0.0000 |
| Fe(III)$^2$ | 0.000114 | 0.000036 | 3.161 | 0.0101 |
| Fe(VI)$^2$ | 0.120680 | 0.042542 | 2.837 | 0.0176 |
| Fe(III):Fe(VI) | 0.006892 | 0.001003 | 6.870 | 0.0000 |

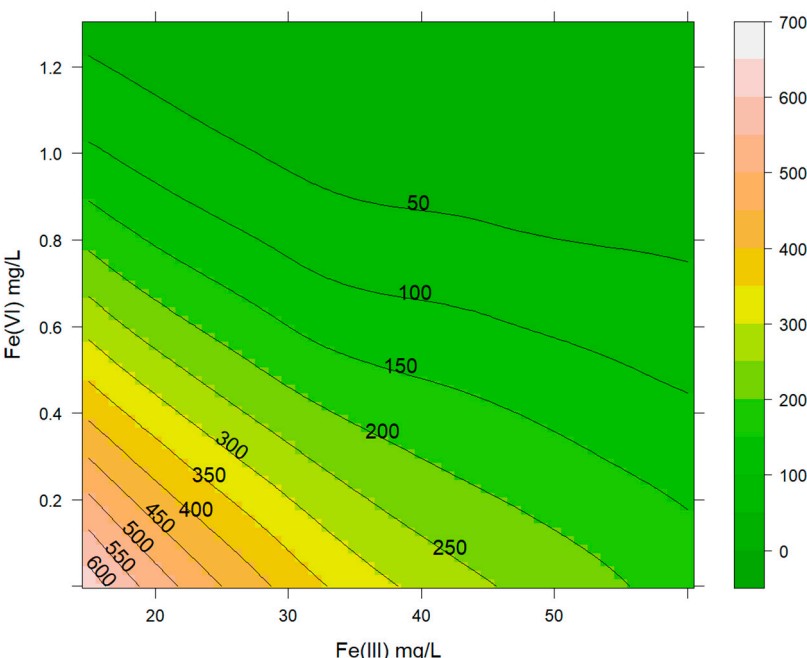

**Figure 1.** Local polynomial regression adjustment to assess the effects of Fe(III) and Fe(VI) on the final arsenic concentrations at the initial As(III) concentration of 1000 µg $L^{-1}$.

The adsorption of As(V) on the iron oxyhydroxides surface (previous to coprecipitation) has been described as a process where an inner sphere surface complex is formed [46] (Equation (6)).

$$Surface - OH + H_3AsO_4 \rightarrow Surface - AsO_4^{-2} + H_2O + 2H^+ \tag{6}$$

Jain et al. [23] working with an initial concentration of 500 µg As(III) $L^{-1}$ obtained 0.7 µg $L^{-1}$ residual arsenic concentration after combined treatment with 0.8 mg $L^{-1}$ of Fe(VI) as an oxidant and 20 mg $L^{-1}$ of Fe(III) as a coagulant, Fe(VI) was not tested during As(V) removal tests.

### 3.2. Effects of Ferrate(VI) Ions Combined with Fe(III) Ions in the Removal of As(V)

Table 3 denotes significant effects only for the linear and quadratic coefficients of Fe(III). The Fe(III) linear coefficient is negative, implying that the concentration of As(V) decreases as the concentration of Fe(III) increases. The coefficients of Fe(VI) are not significant, which is consistent with the fact that As(V) is already oxidized. Hence, the oxidizing function of Fe(VI) is not required, and the extra iron provided by the Fe(VI) is insufficient to exert a clear effect on As(V) removal as shown in Figure 2, where the isolines are nearly vertically

parallel within 0–30-mg L$^{-1}$ range of Fe(III). Fe(VI) has been used to remove efficiently As(V) but required a Fe(VI):As mass ratio of 4:1 [45].

**Table 3.** *t*-test coefficients for a Quadratic Polynomial Regression Model with interaction that explains the final arsenic concentration at the initial As(V) concentration of 1000 µg L$^{-1}$.

| Factors | Estimated Coefficient | Standard Error | *t*-Value | *p*-Value |
|---|---|---|---|---|
| Fe(III) | −0.002775 | 0.000441 | −6.288 | 0.0001 |
| Fe(VI) | 0.011415 | 0.010710 | 1.066 | 0.3116 |
| Fe(III)$^2$ | 0.000030 | 0.000006 | 5.359 | 0.0003 |
| Fe(VI)$^2$ | 0.003330 | 0.006645 | 0.501 | 0.6272 |
| Fe(III):Fe(VI) | −0.000320 | 0.000157 | −2.042 | 0.0685 |

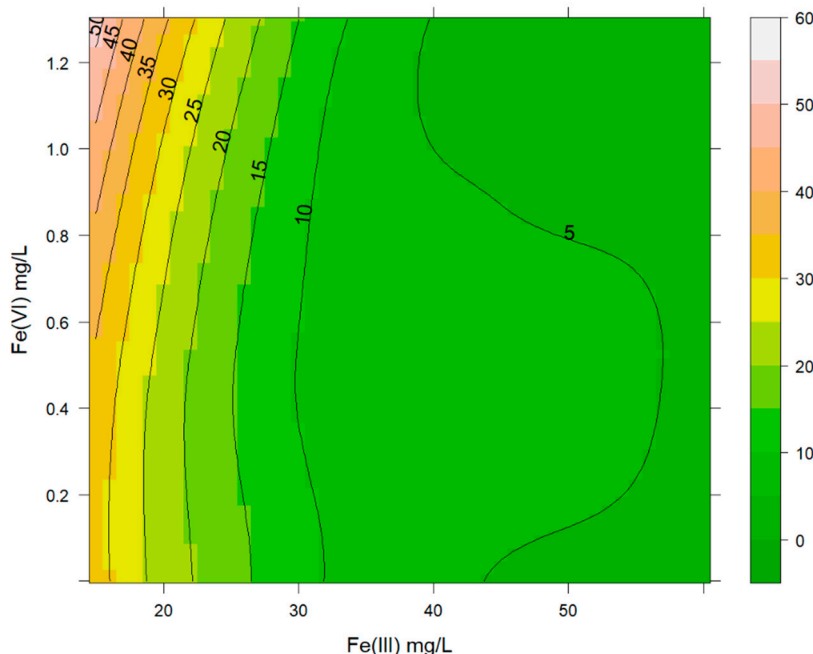

**Figure 2.** Local polynomial regression adjustment to assess the effects of Fe(III) and Fe(VI) on the final arsenic concentrations at the initial As(V) concentration of 1000 µg L$^{-1}$.

*3.3. Effects of Ferrate(VI) Ions Combined with Fe(III) Ions in the Removal of As(III) Combined with As(V)*

For the removal of As(III) combined with As(V), Fe(III) and Fe(VI) exhibit significant linear effects (higher concentration, less remaining arsenic), but only Fe(VI) shows a significant quadratic effect (Table 4). Figure 3 suggests a stronger effect due to Fe(VI), a higher Fe(VI):As(III) ratio results in higher oxidized As(III) proportion, which is translated in lower residual arsenic concentrations than those shown in Figure 1 where the As(III) concentration was 1000 µg L$^{-1}$.

**Table 4.** Local polynomial regression adjustment to assess the effects of Fe(III) and Fe(VI) on the final arsenic concentrations at the initial concentration of 500 µg L$^{-1}$ of As(III) and 500 µg L$^{-1}$ of As(V).

| Factors | Estimated Coefficient | Standard Error | *t*-Value | *p*-Value |
|---|---|---|---|---|
| Fe(III) | −0.008022 | 0.003044 | −2.636 | 0.0249 |
| Fe(VI) | −0.504883 | 0.073853 | −6.836 | 0.0000 |
| Fe(III)$^2$ | 0.000054 | 0.000039 | 1.394 | 0.1936 |
| Fe(VI)$^2$ | 0.199681 | 0.045822 | 4.358 | 0.0014 |
| Fe(III):Fe(VI) | 0.003098 | 0.001081 | 2.867 | 0.0168 |

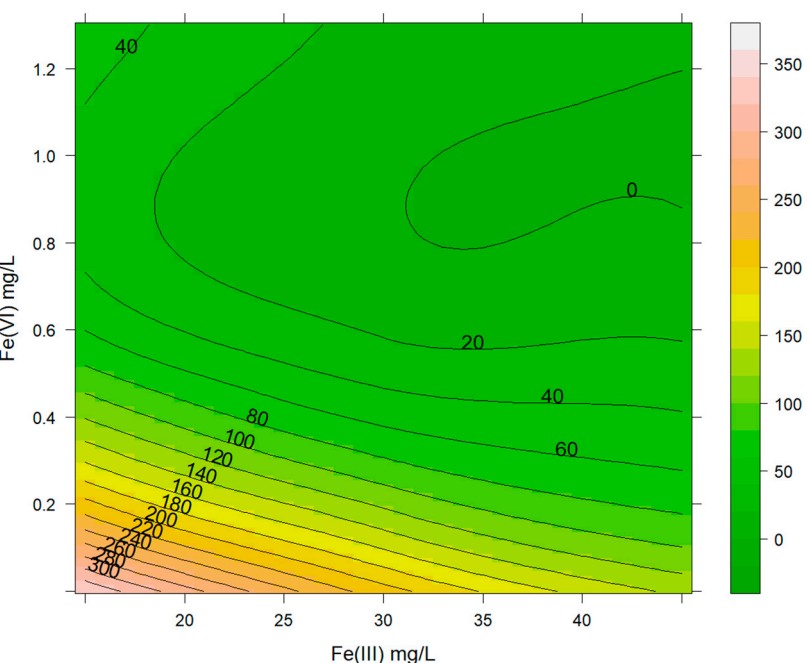

**Figure 3.** Local polynomial regression-fitting adjustment to assess the effects of Fe(III) and Fe(VI) on the final arsenic concentrations at the initial concentration of 500 μg L$^{-1}$ of As(III) and 500 μg L$^{-1}$ of As(V).

The use of Fe(VI) allows an important reduction of Fe(III) to achieve comparable arsenic removals, e.g., according to the estimated values of Figure 3, to remove 99% of the arsenic (10 μg L$^{-1}$ residual concentration) we can use a Fe(VI):Fe(III) mass ratio of 0.85:24 or 0.68:40. In that case, the first ratio is more desirable because we add fewer water treatment chemicals, which traduces in better water quality and less sludge production.

### 3.4. Arsenic Removal Using Different As(III) and As(V) Proportions

The removal of As(III), and of the As(III)/As(V) mixtures by the combined action of Fe(VI) and Fe(III) were effective at a total arsenic concentration of 1000 μg L$^{-1}$. The percentage of arsenic removal by any coprecipitation process depends directly on the initial concentration of arsenic, among other variables [47]. Therefore, arsenic removal effectiveness with a combination of 1 mg L$^{-1}$ of Fe(VI) and 30 mg L$^{-1}$ of Fe(III), identified as a highly effective combination in Figure 3, was tested against different As(III)/As(V) proportions, decreasing the total arsenic concentration in the mixture to 500 μg L$^{-1}$ and 250 μg L$^{-1}$ (Figure 4). Arsenic removal—measured as a total arsenic concentration—is significantly greater (F-test, *p*-value = 0.0196) at 500 μg L$^{-1}$ (99% removal) than at 250 μg L$^{-1}$ (96.9% removal) and similar for all the four As(III):As(V) proportions (no significant differences with an F-test, *p*-value = 0.4447). When the pollutant concentration decreases, the removal mechanisms are faced with mass transfer limitations, and their removal efficiency decreases [48].

The speciation tests also reveal that, in practically all cases, the removal of As(III) was below the detection limit of the equipment (Figure 5) except for the 80:20 As(III):As(V) proportion at a concentration of 250 μg L$^{-1}$.

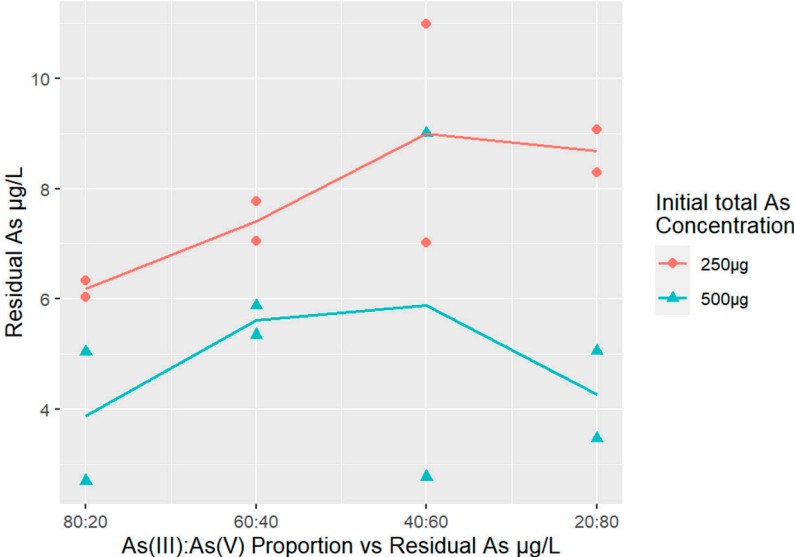

**Figure 4.** Residual arsenic concentration after treating four As(III):As(V) proportions at a total concentration of 250 μg L$^{-1}$ (dots) and 500 μg L$^{-1}$ (triangles) with two repetitions. Lines connect the mean values (6.19, 7.41, 9.00, and 8.68 for a concentration of 250 μg L$^{-1}$ and 3.88, 5.61, 5.89, and 4.27 for a concentration of 500 μg L$^{-1}$). Fe(VI) and Fe(III) dosage of 1 mg L$^{-1}$ and 30 mg L$^{-1}$, respectively.

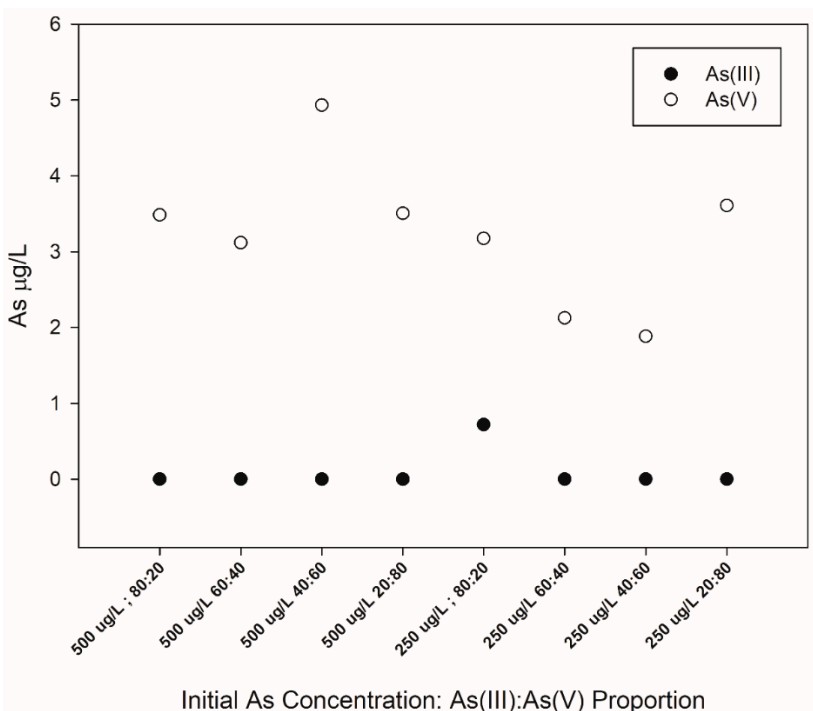

**Figure 5.** Final arsenic concentration per species at different final As(III):As(V) concentrations and proportions.

## 4. Conclusions

Combined treatment using 1 mg Fe(VI) and 30 mg Fe(III) for As(III)/As(V) mixture produced a final average removal of 99.0% and 96.9% for the 500 and 250 μg L$^{-1}$ concentrations, respectively. The arsenic removal percentage -measured as a total arsenic concentration is significantly greater at 500 μg L$^{-1}$ than at 250 μg L$^{-1}$, and is similar for all the As(III)/As(V) tested proportions (no significant differences).

The use of Fe(VI) allows the use of a lower dose of Fe(III) ions to reach the limit of 10 µg L$^{-1}$ treating equimolar mixtures of As(III)/As(V) at a total arsenic concentration of 1000 µg L$^{-1}$. Increasing the dose of Fe(VI) from 0.68 to 0.85 mg L$^{-1}$ allowed the reduction of the Fe(III) dosing from 40 to 24 mg L$^{-1}$.

Regardless of whether As(III) or As(V) was the species present in higher proportion at the beginning of tests, after Fe(VI)/Fe(III) combined treatment the predominant form was As(V) and below the 10 µg L$^{-1}$ threshold. As(V) is less toxic than As(V) but still dangerous; therefore, is worth noting that residual arsenic concentration and species must be determined frequently when applying the described removal process to natural underground waters whose characteristics can fluctuate with time.

**Author Contributions:** Conceptualization, J.Q.-F., R.E.P. and L.F.d.P.; methodology, J.Q.-F., R.E.P.; statistical analysis, R.E.P.; investigation, J.Q.-F., R.E.P., P.P.V., P.M.G.; writing—original draft preparation, J.Q.-F.; L.F.d.P.; writing—review and editing, J.Q.-F., R.E.P. and L.F.d.P. All authors have read and agreed to the published version of the manuscript.

**Funding:** The Instituto de Investigación Científica de la Universidad de Lima (IDIC) provided the funding for this research under contracts 56.017.2014 and 56.012.2015.

**Institutional Review Board Statement:** Not applicable.

**Informed Consent Statement:** Not applicable.

**Data Availability Statement:** The data presented in this study are openly available in FigShare at 10.6084/m9.figshare.14442776.

**Acknowledgments:** The Instituto de Investigación Científica de la Universidad de Lima (IDIC), Laboratorio de Docimasia de la Facultad de Ingeniería Industrial de la Universidad de Lima.

**Conflicts of Interest:** The authors declare no conflict of interest.

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
