# Peer review of "Assessing the Removal of Arsenite and Arsenate Mixtures from the Synthetic Bangladesh Groundwater (SBGW) Using Combined Fe(VI)/Fe(III) Treatments and Local Regression Analysis"

_water, doi:10.3390/w13091134_

Round 1

Reviewer 1 Report

Comments and Suggestions for Authors in Attachment

Author Response

Dear reviewer,

We would like to thank you for your time, feedback, comments and suggestions. We appreciate them. Responses follow.

Comments and Suggestions for Authors
- The novelty of the work is not clearly explained. I would like to suggest to the authors to define better the aim and to point out the novelty of the work. The most important  achievement result from the conducted research with regard to the previously published  results should be emphasized.

We appreciate your comment, we agree.

Section 1, introduction, was modified from line #76-#90.  There we explain that studies on arsenic removal are focused in single species or, if using both species, in separate testing. As natural underground waters are highly probable to contain both arsenic species, the novelty is to test the the proposed treatment on mixtures, evaluate the effects of proportions in this mixtures and do an speciation analysis of the residual arsenic. We also tried to express study objectives in a clearer way.

-Arsenite and As (III) both appear in the text. For the sake of consistency, please choose one format and revise accordingly. The same with arsenate and As(V).

Thank you for the comment. We are using the terms arsenite/arsenate in the title and As(III)/As(V) in the rest of the document.

- Results and discussion. One of the problem with the analysis of the results is that are not compared with the existing literature. Please provide relevant comparisons.

We appreciate your feedback.

Section 3.1 includes two references addressing the aplicability of Fe(VI) for As(V) removal [47] and the result of the use of Fe(VI) and Fe(III) to remove As(III) [23].

Section 3.2 includes reference where it is stated the necessary mass ratio Fe(V):As(V) to remove arsenic [47].

Section 3.3 includes a comparison between two mass Fe(VI)/Fe(III) ratio to obtain similar removals, calculated from the local regression plots (lines 236-240). We did not found literature with a similar development that we could use for comparison.

Section 3.4 we used two references [50] and [51] to explain some of the observed results however, we have not found other works evaluating the removal of diverse ratios between both arsenic species.

-Line 85: Instead of Na2HCO3 it should be NaHCO3.

We appreciate the level of detail in the revision. Chemical formula of bicarbonate was corrected (line 96).

- The Conclusions is recommended to contain more numerical data emphasizing the more important results.

Thank you for the suggestion. Section 4, conclusions, was rewritten to emphasize numerical data (lines 274-282). A conclusion suggested by reviewer 2 was also added (lines 283-288).

- The literature is older, the references from last year's are necessary, for demonstrated the actuality. Please provide references from 2019 and 2020.

We appreciate your feedback, new recent references were added to the study.

2021: [27], [29], [30], [37], [48]

2020: [31], [34] 

2019: [32]

- The manuscript needs significant improvements before its publication. I propose the adoption of the manuscript after major revision

We appreciate the comments and suggestions, manuscript underwent extensive revision.

Reviewer 2 Report

High concentrations of arsenic in drinking water is a huge economic problem. This chemical element accumulates in tissues rich in creatine, ie the epithelium of the gastrointestinal tract, hair, skin, nails. The inorganic forms of arsenic have been classified by the World Organization for Research on Cancer (IARC) into the 1st toxicity group, i.e. as carcinogenic substances.

The methods used to remove arsenic from water are: coagulation, lime precipitation, ion exchange, membrane processes, sorption methods and oxidation of iron and manganese. The authors used a combination of Fe(VI) and Fe(III) to remove arsenic from water. The authors documented that the use of Fe(VI) / Fe(III) effectively removes As(III) and As(V) (at concentrations of 1000, 500 and 250 μg / L) up to the limit established by the WHO for drinking water. The authors showed that the use of ferrate (VI) also allows the use of a much lower dose of ferric ions to reach the limit of 10 μg / l. In controlled laboratory conditions, they selected the most effective combination of Fe(VI) / Fe(III) concentrations for the defined proportions of three and five-valued arsenic species.

In the article, I did not record the chemical reactions of arsenic precipitation from water. Please add them to the text. 

In conclusions, it is worth emphasizing that regardless of whether As (III) or As (V) was the dominant form of arsenic at the beginning, As (V) dominates after reaction with iron. Waters containing predominantly arsenic (V) speciation can be considered less hazardous to the consumer, although it should be remembered that the overall concentration of arsenic in the water and the period in which it was taken are very important.

Author Response

Dear reviewer,

Thank you very much for your time, feedback and comments, we  appreciate them.

Reviewer 2

High concentrations of arsenic in drinking water is a huge economic problem. This chemical element accumulates in tissues rich in creatine, ie the epithelium of the gastrointestinal tract, hair, skin, nails. The inorganic forms of arsenic have been classified by the World Organization for Research on Cancer (IARC) into the 1st toxicity group, i.e. as carcinogenic substances.

The methods used to remove arsenic from water are: coagulation, lime precipitation, ion exchange, membrane processes, sorption methods and oxidation of iron and manganese. The authors used a combination of Fe(VI) and Fe(III) to remove arsenic from water. The authors documented that the use of Fe(VI) / Fe(III) effectively removes As(III) and As(V) (at concentrations of 1000, 500 and 250 μg / L) up to the limit established by the WHO for drinking water.

The authors showed that the use of ferrate (VI) also allows the use of a much lower dose of ferric ions to reach the limit of 10 μg / l. In controlled laboratory conditions, they selected the most effective combination of Fe(VI) / Fe(III) concentrations for the defined proportions of three and five-valued arsenic species.

In the article, I did not record the chemical reactions of arsenic precipitation from water. Please add them to the text.

Thank you for the input, reactions are critical to understand the arsenic removal mechanism. 

Reactions (lines191-201) depicting arsenate oxidation (equation 3), hydrolysis of ferric ions (equation 4), precipitation (equation 5) and adsorption-coprecipitation (equation 6) as well references supporting the proposed reactions [48] and [49] added in section 3.1

We also generated a comparison of Fe(VI):Fe(III) mass ratios required to reach 10 µg/L residual arsenic in the lines 235-238.  Thank you for pointing out it (lines 236-240).

In conclusions, it is worth emphasizing that regardless of whether As (III) or As (V) was the dominant form of arsenic at the beginning, As (V) dominates after reaction with iron. Waters containing predominantly arsenic (V) speciation can be considered less hazardous to the consumer, although it should be remembered that the overall concentration of arsenic in the water and the period in which it was taken are very important.

Thank you for the comment, it is important and we did not emphasized it.  Conclusion regarding predominance of As(V) post-treatment, advantages, risk and cautions added to section 4 (lines 283-288) which was rewrited to emphasize numerical results.

Round 2

Reviewer 1 Report

Comments and Suggestions for Authors in attachment
